# Corporate social responsibility and cross-border M&A: The moderating effect of institutional distance

**Haiting Li[1], Shuzhen Li[2]\*, Xiangcen Zhan[3], Feng Zhang[2], Mingwei Sun[4]**

**1** School of Economics and Management, Yantai University, Yantai, China, **2** School of Economics, Nankai University Binhai College, Tianjin, China, **3** School of Economics, Nankai University, Tianjin, China, **4** School of Economics, Zhejiang University, Hangzhou, China

\* lishuzhen1122@126.com

## Abstract

Drawing upon a dataset of cross-border mergers and acquisitions (M&A) events of Chinese enterprises from 2010 to 2017, this study investigates the impact of corporate social responsibility (CSR) on the completion of cross-border M&A with a focus on the moderating role of institutional distance. The results highlight the significance of CSR on the completion of cross-border M&A. The robustness tests including changing estimation model, new measurements, propensity score matching, and instrumental variable tests show that the main results are consistent. Second, both formal and informal institutional distance have positive moderating effects of CSR on the completion of cross-border M&A.

## Introduction

Cross-border mergers and acquisitions (M&A), also known as overseas M&A, refers to the corporate action of purchasing the shares or assets of another company in a foreign country [1]. The past 30 years witnessed a boom in cross-border M&A, with various countries and industries participating in global acquisitions [2]. Issues surrounding corporate social responsibility (CSR) draw increasing attention from numerous scholars, though few empirical studies focus on the role of CSR in M&A, especially in the realm of cross-border M&A. CSR is about business and other organizations going beyond the legal obligations to manage the impact they have on the environment and society. In particular, this could include how organizations interact with their employees, suppliers, customers and the communities in which they operate, as well as the extent they attempt to protect the environment [3].

Stakeholder theory and shareholder theory offer two conflicting views of CSR in M&A. From the stakeholder theory perspective, an enterprise should not only consider the interests of shareholders, but also place equal weight on the interests of other stakeholders. Enterprises presenting a high level of social responsibility will be more likely to be recognized by the merged or acquired enterprise, thereby facilitating the M&A process [4]. However, from the shareholder view, high CSR usually means that enterprises overemphasize the interests of

**Data Availability Statement:** All the financial data corresponding the control variables are from the CNRDS database and China Stock Market & Accounting Research (CSMAR www.cnrds.com. https://www.gtarsc.com/ We collected CSR data

from Hexun.com. We calculated the formal institutional distance using data from the Worldwide Governance Indicators (WGI) database and measured informal institutional distance using Geert Hofstede's data.

**Funding:** SZL 1 Supported by Research Development Program of Tianjin Municipal Education Commission Grant No. 2020SK134:" Research on the Impact of Environmental Control Policies on Corporate Governance Mechanism." The funder had no role in study design, data collection and analysis, decision to publish, or preparation of the manuscript.

**Competing interests:** The authors have declared that no competing interests exist.

other stakeholders over the interests of shareholders, hence shareholders may not be willing to merge with companies with high CSR.

Thus, further studies are required to resolve the inconsistencies of prior findings. Specifically, this study chooses China as the research context for two reasons. First, according to the statistics from open sources, the money spent on cross-border M&A transactions by Chinese companies greatly increased, from 34.7 billion dollars in 2010 to 150.6 billion dollars in 2016, and the number of transactions escalated from 268 to 738. Furthermore, with the reinforcement of China's "Go Global" strategy and the Belt and Road initiative, as embodied in the upgrading of corporations, the process of cross-border M&A by Chinese enterprises is growing significantly. For example, Chinese companies investing in these countries conducted at least 192 M&A transactions in 2016, representing an aggregate value of 19.3 billion dollars, in contrast with 46 transactions and 4.6 billion dollars in 2010. However, the completion rate of cross-border M&A by Chinese enterprises seems to be below the world average. This outcome may indicate that Chinese enterprises may not notice the underlying factors that may lead to failures in their cross-border M&A activities.

Second, China, the largest emerging market with the transition into the market economy, is integrating into the world economy and enhancing the governance quality, thereby attracting increasing academic attention. Considering that there are many common characteristics between China and other emerging markets, this study advances our knowledge of emerging markets. Additionally, it may contribute to the stream of literature on M&As by investigating how firms from emerging markets complete cross-border M&A deals, especially considering that firms from these regions experience a high percentage collapse before completion [5].

We commonly see CSR in emerging markets, as Chinese enterprises also recognize the advantages of CSR practices. CSR enhances value, corporate reputation, foreign investment, and access to markets in developed economies [4]. The significance of CSR is in its contributing to a vital dynamic in the social and economic development in China, though prior studies do not provide conclusive data on this "effect". Therefore, this study investigates the role of CSR on the completion of cross-border M&A by Chinese enterprises using a sample of cross-border M&A events of these enterprises from 2010 to 2017. To resolve the inconsistences between the shareholder and stakeholder views, we further examine the effect of CSR contingent on institutional distance. Introducing the contextual factors of institutional distance enables us to clarify the underlying role of CSR. To the best of our knowledge, this is the first empirical study to examine the interaction of CSR and institutional distance on the completion of cross-border M&A in emerging markets.

First, we find that CSR has a positive role in the completion of the cross-border M&A of Chinese enterprises, thereby supporting the stakeholder theory. Second, both formal and informal institutional distance moderate the effect of CSR positively. The results of this study can help increase the completion rate of cross-border M&A by Chinese enterprises as it investigates the dynamic nature between institutional distance and CSR in cross-border M&A.

## Literature review and hypothesis development

### CSR and cross-border M&A

While several works present diverse perspectives on the benefits of CSR, the empirical findings are inconsistent. Some studies report a positive relationship between a company's CSR and its financial performance [6–8]. Multinational corporations' (MNCs') compliance with environmental and CSR standards may be driven by customer preferences, customer monitoring, and expected sanctions [9, 10]. Calveras [11] suggests that CSR can act as an instructive device to distinguish product quality. Some research indicates that firms can signal their orientation to

consumers through high expenditure on CSR activities, which may be positively or negatively correlated with profits. The correlation depends on the need for CSR, the firm's ability to align with CSR, and the degree of competition in the market [12]. Wong [13] posits that CSR acts as an external driver that has a strategic influence on local institutional order, rendering local systems more transparent and amendable to changes. Based on a large sample of US firms, firms with better CSR scores have less expensive equity financing [14]. According to Cho et al. [15], CSR performance has a positive role for investors as it reduces information asymmetry. In addition, firms' CSR performance is negatively associated with future crash risk [16]. Rahman et al. [17] find a positive correlation between CSR activities and marketing performance; customers view CSR activities positively and tend to reward such activities by purchasing more products and services from these companies. In some cases, CSR is treated as an accoutrement to differentiate firms in a competitive market [18]. In times of economic crisis, socially responsible strategies are a determinant factor in small firms' competitiveness [19]. Zaman et al. [20] suggest that CSR, affective organizational commitment, and organizational identification are positively correlated, all of which can help businesses achieve superior performance and sustainable success.

In terms of the impact of CSR on cross-border M&A, using the stakeholder theory, Hawn [21] reports that the successful completion of cross-border M&A depends largely on stakeholders' evaluation of the merged and acquired firm. In his study, emerging market multinationals (EMMs) that tend to have adverse news with respect to their CSR engagement at home are less likely to complete cross-border deals and more likely to take longer than their counterparts. Bereskin et al. [22] show that firms sharing similar CSR profiles are more likely to manage a merger, conclude their deals more quickly, enjoy greater merger synergies, bolster their long-run performance, and undergo fewer changes in CSR policies after the deal is finalized. A study using a large sample of mergers in the US reports that mergers by high CSR acquirers take less time to complete and are less likely to fail than mergers by low CSR acquirers [23]. Further, Arouri et al. [24] explicitly state that M&A completion uncertainty is negatively related to the CSR of the acquirer.

Conversely, from the shareholder theory perspective, higher CSR implies that companies may weigh the interests of stakeholders against those of shareholders, so shareholders who value their own interests may be unwilling to merge with companies with high CSR. Levitt [25] believes the only responsibilities of businesses are "to obey the elementary canons of everyday face-to-face civility and to seek material gain." Advocates of the shareholder view suggest that CSR-related activities benefit other stakeholders at the expense of shareholders [24]. In addition, strong CSR attributes should reduce the probability of a breach in implicit contracts and therefore increase stakeholders' support of a firm [24]. It is plausible for enterprises that lack internal CSR to either circumvent the public's attention on inappropriate behaviors or misguide the public by deliberately advertising high CSR behaviors. In fact, this kind of behavior damages the interests of stakeholders, providing a disreputable signal of the acquired enterprise, and thus grinding the M&A to a halt.

Specifically, we suppose that CSR has a positive effect on the completion of M&A for Chinese companies. The emerging market is characterized by institutions that are transitioning to the market economy, with their implementation still likely to be erratic and their norms continuing to evolve [26]. Consequently, firms from emerging markets like China may be likely to suffer the liability of foreignness when conducting cross-border M&A in international markets. Accordingly, as a strategic tactic, CSR enables these firms to enhance their legitimacy in host countries and build trustworthiness with the target firms' stakeholders [27], thereby overcoming the liability of foreignness [28] and facilitating the completion of M&A. Therefore, we suppose that,

*Hypothesis 1*: *CSR has a positive effect on cross-border M&A.*

## Formal and informal institutional distance

Institutions are "stable, values, recurring patterns of behavior" [29, 30] that reflect the "rules and routines that define actions in terms of relations between roles and situations" [29, 31]; that is, institutions represent the rules of a society [32]. Institutions reduce uncertainty by creating a stable framework for people and organizations to interact.

Institutional distance is an indicator of the difference between two national systems. Differences in national formal and informal institutions can explain part of the variation in the likelihood that an announced cross-border acquisition deal will be completed and can reduce the duration of the deal-making process [33]. Berry et al. [34] illustrate how institutional distance can have differing effects on firms' decisions when engaging in foreign investment. Zademach and Rodríguez-Pose [35] show that the traditional motives of entering new major markets, the effects of geographical adjacency, and the incorporation of localized capabilities represent the key drivers of European M&As, while institutional factors, such as European integration or language barriers, appear to be less influential. The institutional theory postulates that MNCs must appease various institutional pressures to establish legitimacy in the host nation, which in turn facilitates their business success and continuous market survival [33]. Hur et al. [36] indicate that the difference in the quality of institutions between developed and developing countries affect cross-border M&A inflows. Compared with China-Japan M&As, US-Japan M&As reduce the growth of the stock prices of Japanese targets [37].

Specifically, we suppose that institutional distance raises the risk and uncertainty of firms' internationalization and M&A [38], while CSR may reduce the uncertainties [24], thereby enhancing the success of M&A. Firms looking to invest overseas face a natural disadvantage when they conduct multinational operations as "outsiders", especially when institutional distance is high. Enterprises entering a host country's market mainly face the problem of local unfamiliarity caused by information asymmetry, along with discrimination due to the lack of legitimacy and relationship harm as they lack embeddedness. First, the unfamiliarity hazard mainly refers to the fact that overseas enterprises entering a host country's market are at a disadvantage in understanding the host country's market situation, legal system, and culture. Obtaining such information often requires a relatively high cost and high engagement in CSR to gain trust and recognition among the host countries' stakeholders. For example, Kostova and Zaheer [38] find that when the host and home countries have high institutional distance, CSR is more attractive for foreign firms, and the more positive its CSR activities, the easier it is for them to correctly understand the host country's institutional environment, especially the informal implicit social norms, cultural practices, religious beliefs, and conventional rules.

Second, the harm caused by the lack of legitimacy refers to the lack of information to judge a foreign firm's operations in the host country, where locals often rely on stereotypes to judge the behavior of the company. If the foreign firm has higher engagement in positive CSR activities or show successful investment behaviors, then the host country's stakeholders will believe that the most obvious characteristics are positive and reliable.

Finally, the damage to relationships caused by the lack of embeddedness mainly refers to the fact that a new foreign firm does not have effective and close relationships with the host country government, suppliers, consumers, communities, and other stakeholders. Thus, firms establish relations with these parties by engaging in various CSR activities to achieve cross-border M&A. Therefore, we examine the moderating effect of formal and informal institutional distance individually. Our research model in Fig 1 summarizes the framework of this study.

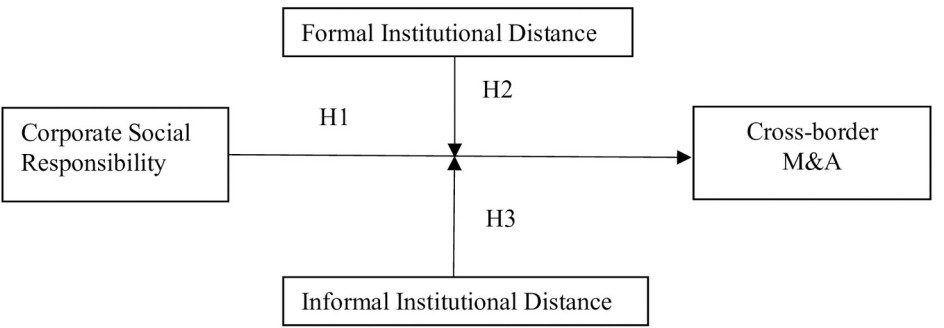

**Fig 1. The conceptual framework.**

*Hypothesis 2: Formal institution positively moderates the effect of CSR on cross-border M&A.*

*Hypothesis 3: Informal institution positively moderates the effect of CSR on cross-border M&A.*

## Data and methodology

### Data sources

We draw upon a dataset of cross-border M&A events of Chinese A-share non-ST listed enterprises from 2010 to 2017 from the China Research Data Service Platform (CNRDS database) to examine the impact of their CSR on cross-border M&A. All the financial data corresponding to the control variables are from the CNRDS and the China Stock Market & Accounting Research (CSMAR) databases. We collected CSR data from Hexun.com. Founded in 1996, Hexun.com built the first vertical financial portal website in China and developed as an authorized financial and securities information provider. It provides CSR score data for all listed companies according to CSR reports released on the official websites or in the annual reports issued by listed companies. The CSR observed value provided by Hexun.com contains five aspects, with measurements similar to those of KLD data employed in prior studies [39]: shareholder responsibility, employee responsibility, supplier and customer responsibility, environmental responsibility, and social responsibility. Among them, there are 13 secondary indicators and 37 tertiary indicators.

### Model

To investigate the impact of CSR on cross-border M&A, we establish the following basic model:

$$Completion = \beta_0 + \beta_1 \times CSR + \sum controls + \varepsilon \tag{1}$$

With this foundation, we use formal and informal institutional distance as the moderating variables to construct the following new models respectively:

$$Completion = \beta_0 + \beta_1 \times CSR + \beta_2 \times Formal\ Institutional\ Distance + \beta_3 \times CSR$$

$$\times Formal\ Institutional\ Distance + \sum controls + \varepsilon \tag{2}$$

$$Completion = \beta_0 + \beta_1 \times CSR + \beta_2 \times Informal\ Institutional\ Distance + \beta_3 \times CSR$$

$$\times Informal\ Institutional\ Distance + \sum controls + \varepsilon \tag{3}$$

**Dependent and independent variables.** The dependent variable is whether the cross-border M&A was completed or not (*Completion*). *Completion* is a binary variable [33] equal to 1 if the cross-border M&A was completed, and 0 otherwise. The independent variable is CSR. Prior research observes CSR according to the following 13 categories: community, diversity, employment, environment, human rights, product, alcohol, gaming, firearms, military, nuclear, tobacco, and corporate governance [17]. As we use Hexun.com data, our CSR measurement consists of five areas: shareholder responsibility; employee responsibility; supplier, customer, and consumer responsibility; environmental responsibility; and social responsibility. These areas include the most significant categories of community and environment activities [17, 40, 41] and incorporates stakeholder responsibility. For this study, we measure CSR by taking the natural logarithm of the total score obtained by aggregating the five sections.

**Moderating variables.** The two moderating variables in this study are *Formal Institutional Distance (FID)* and *Informal Institutional Distance (IID)*.

By employing data from the Worldwide Governance Indicators (WGI) database, this study calculated formal institutional distance. The WGI project measures the development of a country's institutions by using six indicators: (1) Voice and Accountability; (2) Political Stability and Absence of Violence; (3) Government Effectiveness; (4) Regulatory Quality; (5) Rule of Law; and (6) Control of Corruption. Following the method of Li et al. [42], formal institutional distance is calculated in the Euclidean distance. Here, $FID_j$ represents formal institutional distance between the host country j and the home country, $I_{nj}$ is the score of dimension n for the host country in year t, $I_{nc}$ denotes the score of dimension n for the home country in the year t, and $V_n$ denotes the variance of dimension n in the same year.

$$FID_j = \sqrt{\sum_n^6 \frac{(I_{nj} - I_{nc})^2}{V_n}} \tag{4}$$

North [32] argues that customs, traditions, and codes of conduct carry informal constraints. As in existing studies that evaluate informal institutional distance [43], our measurement of informal institutional distance also focuses on cultural distance. We measure informal institutional distance by using the index of uncertainty avoidance in Hofstede's national culture: (1) Individualism; (2) Power Distance; (3) Masculinity; (4) Uncertainty Avoidance; (5) Long-term Orientation; and (6) Indulgence. We also use Euclidean distance to calculate informal institutional distance.

$$IID_j = \sqrt{\sum_m^6 \frac{(I_{mj} - I_{mc})^2}{V_m}} \tag{5}$$

**Control variables.** First, to reduce the concerns related to model misspecification, especially the omitted important variables, we generated year, industry, and region dummies, along with firm dummies, and added them into the model, so that we could rigorously control for all other unobservable variables [42, 43].

Second, at the merging and acquiring enterprise level, this paper controls for the following variables: (1) Firm age (Ln_Age) is the natural logarithm of the number of years since the merging and acquiring company started operating. (2) Firm size (Ln_Assets) is the natural logarithm of the total assets of the merging and acquiring company at the end of the year [44]. (3) Cash flow (Cash_ratio) is the ratio of the sum of cash and cash equivalent to current liabilities, which reflects the operation of the enterprise. (4) Working capital ratio (Flow_ratio) is the

**Table 1. Descriptive statistics.**

| Variable | Obs | Mean | Std. Dev. | Min | Max |
|---|---|---|---|---|---|
| Completion | 639 | 0.448 | 0.498 | 0 | 1 |
| CSR | 631 | 3.293 | 0.664 | -2.408 | 4.452 |
| Formal Institutional Distance | 412 | 3.972 | 1.548 | 0 | 5.931 |
| Informal Institutional Distance | 344 | 3.137 | 1.532 | 0.304 | 5.419 |
| Ln_age | 639 | 2.797 | 0.348 | 1.099 | 3.526 |
| Ln_assets | 562 | 23.310 | 2.364 | 19.505 | 30.815 |
| Cash_ratio | 595 | 0.861 | 1.874 | 0.015 | 16.440 |
| Flow_ratio | 595 | 2.279 | 3.104 | 0.159 | 42.479 |
| Asset_liability_ratio | 638 | 0.496 | 0.224 | 0.033 | 1.099 |
| SOE | 549 | 0.362 | 0.481 | 0 | 1 |
| Competition | 489 | 1.014 | 0.135 | 1 | 3 |

ratio of current assets to current liabilities measuring the risk of operation, which is vital for cross-border M&A. (5) Asset_liability_ratio is the ratio of total liabilities to total assets, which is a measurement of financial leverage [45]. (6) Equity ownership (SOE) is a binary variable equal to 1 if the acquiring company is a state-owned enterprise, and 0 otherwise. State ownership ties are likely to influence the regulated resources and policy treatment that a firm can acquire from the government [46, 47], which may affect the competition of an M&A event. (7) Competition is measured by the number of competing buyers for overseas M&A.

Table 1 provides the descriptive statistics of each variable in this study. Furthermore, we present year-wise classifications for completed and non-completed cross-border M&A deals, a distribution of target countries between these two groups, summary statistics of all variables classified according to the sample mean and median of the CSR scores and the test results for the significance of the differences for a better understanding of our sample distribution. These results are provided in Tables 2, 3 and 4 respectively. As shown in Table 4, firms completing cross-border M&A deals have higher CSR scores than firms with failed overseas M&As, suggesting that firms with higher CSR are more likely to succeed in cross-border M&A deals. Furthermore, Table 5 shows the correlation coefficient matrix of each variable, indicating that there is no strong correlation between variables.

## Results

### Impact of CSR on cross-border M&As

We examine the impact of CSR on the completion of cross-border M&A using a logit model to test the hypotheses with the dependent, independent, and control variables defined in the

**Table 2. Sample distribution by year.**

| Year | Completion = 0 | Completion = 1 | Total |
|---|---|---|---|
| 2010 | 17 | 14 | 31 |
| 2011 | 19 | 16 | 35 |
| 2012 | 28 | 16 | 44 |
| 2013 | 27 | 25 | 52 |
| 2014 | 58 | 52 | 110 |
| 2015 | 103 | 71 | 174 |
| 2016 | 61 | 55 | 116 |
| 2017 | 40 | 37 | 77 |
| Total | 353 | 286 | 639 |

**Table 3. Sample distribution by target countries.**

| Target Country | Completion = 0 | Completion = 1 | Total |
|---|---|---|---|
| America | 44 | 39 | 83 |
| Argentina | 1 | 3 | 4 |
| Australia | 21 | 18 | 39 |
| Austria | 1 | 3 | 4 |
| Brazil | 5 | 3 | 8 |
| Canada | 9 | 9 | 18 |
| Denmark | 3 | 1 | 4 |
| Finland | 2 | 2 | 4 |
| France | 4 | 6 | 10 |
| German | 12 | 25 | 37 |
| India | 2 | 3 | 5 |
| Indonesia | 5 | 0 | 5 |
| Italy | 10 | 10 | 20 |
| Japan | 4 | 8 | 12 |
| Malaysia | 5 | 1 | 6 |
| Netherlands | 4 | 2 | 6 |
| New Zealand | 4 | 3 | 7 |
| Norway | 3 | 1 | 4 |
| Pakistan | 1 | 0 | 1 |
| Poland | 1 | 2 | 3 |
| Singapore | 9 | 6 | 15 |
| Spain | 1 | 7 | 8 |
| Switzerland | 1 | 3 | 4 |
| Thailand | 6 | 2 | 8 |
| Vietnam | 1 | 1 | 2 |
| Uzbekistan | 1 | 0 | 1 |
| Israel | 3 | 3 | 6 |
| Russia | 1 | 1 | 2 |
| Croatia | 0 | 2 | 2 |
| Guinea | 0 | 1 | 1 |
| Congo | 1 | 1 | 2 |
| the Democratic Republic of Congo | 1 | 0 | 1 |
| Libby | 0 | 1 | 1 |
| Gabon | 1 | 4 | 5 |
| Hungary | 0 | 1 | 1 |
| South Africa | 5 | 1 | 6 |
| Botswana | 1 | 0 | 1 |
| Qatar | 1 | 0 | 1 |
| Luxembourg | 1 | 1 | 2 |
| Kazakhstan | 3 | 2 | 5 |
| Columbia | 1 | 0 | 1 |
| Turkey | 0 | 2 | 2 |
| Tanzania | 1 | 0 | 1 |
| Tajikistan | 2 | 1 | 3 |
| Mexico | 0 | 1 | 1 |
| Cayman Islands | 1 | 1 | 2 |
| Czech | 0 | 1 | 1 |

(*Continued*)

**Table 3.** (Continued)

| Target Country | Completion = 0 | Completion = 1 | Total |
|---|---|---|---|
| Slovakia | 0 | 1 | 1 |
| Chile | 1 | 0 | 1 |
| Cambodia | 2 | 2 | 4 |
| Belgium | 1 | 0 | 1 |
| Mauritius | 1 | 0 | 1 |
| Trinidad and Tobago | 1 | 1 | 2 |
| Bolivia | 0 | 2 | 2 |
| Sweden | 3 | 2 | 5 |
| Peru | 1 | 0 | 1 |
| England | 10 | 11 | 21 |
| British Virgin Islands | 5 | 1 | 6 |
| Mozambique | 1 | 1 | 2 |
| the Philippines | 1 | 0 | 1 |
| Portugal | 2 | 2 | 4 |
| Oman | 1 | 0 | 1 |
| Korea | 5 | 3 | 8 |
| Malta | 1 | 3 | 4 |
| Others | 89 | 120 | 209 |
| Total | 308 | 331 | 639 |

previous section. The dependent variable is *Completion*, and the *independent variable* is corporate social responsibility (*CSR*). The control variables are firm age (*Ln_Age*), firm size (*Ln_Assets*), cash flow (*Cash_ratio*), working capital ratio (*Flow_ratio*), *asset_liability_ratio*, equity ownership (*SOE*) and *competition*.

Table 6 reports the estimated logit model results based on the revised model specification. Model (1) is the baseline regression. Model (2) reports the regression results with control variables. In Model (2), the regression coefficients of controls such as Ln_Assets, Flow_ratio and Asset_liability_ratio are significantly positive, denoting that the good financial performances of the acquiring enterprises could motivate overseas M&A. The regression coefficient of CSR is significantly positive, indicating that CSR has a significant positive relationship with Chinese enterprises' cross-border M&A. In other words, the stronger the CSR, the more likely the success of the cross-border M&A, which verifies hypothesis 1. A higher merged enterprise's CSR

**Table 4. Summary statistics by completion.**

| Variables | Completion = 0 | Mean0 | Completion = 1 | Mean1 | Test of Difference |
|---|---|---|---|---|---|
| CSR | 347 | 3.273 | 284 | 3.317 | -0.044 |
| Formal Institutional Distance | 205 | 3.944 | 207 | 4 | -0.056 |
| Informal Institutional Distance | 175 | 3.046 | 169 | 3.230 | -0.184 |
| Ln_age | 353 | 2.769 | 286 | 2.830 | -0.061** |
| Ln_assets | 313 | 23.18 | 249 | 23.48 | -0.304 |
| Cash_ratio | 332 | 0.847 | 263 | 0.878 | -0.031 |
| Flow_ratio | 332 | 2.297 | 263 | 2.256 | 0.041 |
| Asset_liability_ratio | 352 | 0.484 | 286 | 0.511 | -0.027 |
| SOE | 305 | 0.344 | 244 | 0.385 | -0.041 |
| Competition | 275 | 1.018 | 214 | 1.009 | 0.009 |

**Table 5. Correlation coefficients.**

| Variables | (1) | (2) | (3) | (4) | (5) | (6) | (7) | (8) | (9) | (10) | (11) |
|---|---|---|---|---|---|---|---|---|---|---|---|
| **(1) Completion** | 1.000 | | | | | | | | | | |
| **(2) CSR** | 0.0331* | 1.000 | | | | | | | | | |
| **(3) Formal Institutional Distance** | 0.0182 | -0.0598* | 1.000 | | | | | | | | |
| **(4) Informal Institutional Distance** | 0.0601* | 0.0211 | 0.7268* | 1.000 | | | | | | | |
| **(5) Ln_age** | 0.0869* | 0.0331* | -0.0690* | -0.1350* | 1.000 | | | | | | |
| **(6) Ln_assets** | 0.0640* | 0.4177* | -0.0638* | 0.1245* | 0.1751* | 1.000 | | | | | |
| **(7) Cash_ratio** | 0.0082 | -0.0307 | 0.1265* | 0.0923* | -0.0899* | -0.2859* | 1.000 | | | | |
| **(8) Flow_ratio** | -0.0066 | -0.0739* | 0.1208* | 0.1116* | -0.0937* | -0.3374* | 0.8887* | 1.000 | | | |
| **(9) Asset_liability_ratio** | 0.0598* | 0.2120* | -0.0788* | 0.0099 | 0.1967* | 0.6785* | -0.5066* | -0.5664* | 1.000 | | |
| **(10) SOE** | 0.0424* | 0.2727* | -0.0408* | 0.1214* | 0.0633* | 0.6064* | -0.1477* | -0.2008* | 0.4616* | 1.000 | |
| **(11) Competition** | -0.0325* | -0.0358* | 0.0485* | 0.0724* | -0.0494* | 0.0411* | -0.0348* | -0.0423* | 0.0560* | 0.0423* | 1.000 |

Note:

* $p < 0.05$.

level enhances the recognition of the stakeholders of the acquirer. The analytical result is consistent with the stakeholder theory and Hawn's [21] conclusions.

## The moderating effects of formal and informal institutional distance

Based on the positive effect of CSR on cross-border M&As, we add formal institutional distance, the interaction term between formal institutional distance and CSR, and informal institutional distance and the interaction term between informal institutional distance and CSR into the regression model, to explore the moderating effects of formal and informal institutional distance. The regression results are reported in Model (3) and Model (4) of Table 6, respectively.

As reported in Model (3) of Table 6, the interaction term between formal institutional distance and CSR is significantly positive at the 10% confidence level. That is, in the presence of formal institutional distance, companies with high CSR have stronger incentives to conduct M&A activities, supporting Hypothesis 2. Model (4) of Table 6 examines the moderating effect of informal institutional distance, and the interaction term is significantly positive at the 10% confidence level. In other words, the greater the informal institutional distance, the stronger the impact of CSR on the completion of cross-border M&A. Hence, informal institutional distance also positively moderates the positive impact of CSR on the completion of cross-border M&A, supporting Hypothesis 3.

## Robustness test

We replace the Logit model with a Probit model to verify the robustness of the results. As shown in Model (1) of Table 7, the regression results remain robust after changing the model. We also change the measures of CSR with Ln_RKS, which is measured by the logarithm of the score of RKS CSR rankings for listed companies. Evidently, the result in Model (2) is still consistent with the main results. Further, to resolve potential endogeneity problems resulting from omitted variables, we use the Propensity Score Matching (PSM) method to run the regression. Specifically, we divide the sample into two groups based on industry-mean CSR: a treatment group where companies have higher CSR scores than the mean in their respective industries and a comparison group with lower CSR. Then, we apply 1-to-1 nearest neighbor matching method within caliper to balance the observable individual characteristics between these two groups according to variables that influence the possibility of entering the treatment

**Table 6. Main effects and moderating effects.**

|  | (1) | (2) | (3) | (4) |
|---|---|---|---|---|
|  | Completion | Completion | Completion | Completion |
| **CSR** | 1.221*** | 1.299** | 0.734 | -4.721 |
|  | (0.467) | (0.655) | (1.238) | (2.876) |
| **Formal Institutional Distance** |  |  | 0.453 |  |
|  |  |  | (0.449) |  |
| **CSR* Formal Institution Distance** |  |  | 1.273* |  |
|  |  |  | (0.705) |  |
| **Informal Institutional Distance** |  |  |  | 0.433 |
|  |  |  |  | (0.482) |
| **CSR* Informal Institution Distance** |  |  |  | 2.902* |
|  |  |  |  | (1.679) |
| **Ln_Age** |  | 9.858 | 16.094 | 15.585 |
|  |  | (9.406) | (15.202) | (15.698) |
| **Ln_Assets** |  | 2.694* | 5.473** | 3.763 |
|  |  | (1.422) | (2.452) | (3.910) |
| **Cash_ratio** |  | -1.209 | -1.883 | 1.228 |
|  |  | (1.149) | (2.432) | (4.190) |
| **Flow_ratio** |  | 1.220* | 2.288 | -0.305 |
|  |  | (0.688) | (1.623) | (1.339) |
| **Asset_liability_ratio** |  | 15.570*** | 13.052* | -0.668 |
|  |  | (5.192) | (7.823) | (16.162) |
| **SOE** |  | -0.014 | 1.524 | 0.960 |
|  |  | (2.018) | (2.503) | (6.546) |
| **Competition** |  | 1.267 | 1.229 | 1.080 |
|  |  | (1.663) | (2.823) | (2.796) |
| **Year Dummies** | YES | YES | YES | YES |
| **Firm Dummies** | YES | YES | YES | YES |
| **Industry Dummies** | YES | YES | YES | YES |
| **Province Dummies** | YES | YES | YES | YES |
| $R^2$ | 0.1914 | 0.2577 | 0.2836 | 0.3192 |

t statistics in parentheses.

* $p < 0.1$,

** $p < 0.05$,

*** $p < 0.01$.

group. After matching, we estimate the model using the matched sample. The result in column (3) of Table 7 provides evidence consistent with the main conclusion, thus supporting our main analysis. Moreover, we use an instrumental variable (IV) probit analysis. Drawing upon prior studies [14, 48–50], we use industry-mean CSR as the instrumental variable for CSR. Industry-mean CSR is correlated to firm-level CSR as the intensity of CSR from peer companies in the same industry motivates companies to engage more in CSR activities, and it is not linked to cross-border M&A for a particular target [50]. The result is reported in columns (4) in Table 7, showing the robustness of our results.

Further, we conducted the robustness tests for the moderating effects by changing the measurement and calculation of institutional distance, and still found consistent results. For example, we used the absolute value to measure institutional distance, as displayed in Model (1)-(2)

**Table 7. Robustness, PSM and IV tests of main effect.**

| | (1) | (2) | (3) | (4) |
|---|---|---|---|---|
| | | | PSM | IV |
| | Completion | Completion | Completion | Completion |
| CSR | 0.774** | | 1.299** | 2.662*** |
| | (0.369) | | (0.655) | (0.251) |
| Ln_RKS | | 6.888** | | |
| | | (3.091) | | |
| Ln_Age | 5.478 | 10.156 | 9.858 | -0.579 |
| | (5.335) | (23.369) | (9.406) | (0.431) |
| Ln_Assets | 1.588** | 6.890 | 2.694* | -0.415*** |
| | (0.781) | (4.230) | (1.422) | (0.151) |
| Cash_ratio | -0.750 | -14.311** | -1.209 | 0.089 |
| | (0.649) | (6.470) | (1.149) | (0.143) |
| Flow_ratio | 0.749* | 9.392 | 1.220* | -0.032 |
| | (0.398) | (6.171) | (0.688) | (0.095) |
| Asset_liability_ratio | 9.466*** | 11.903 | 15.570*** | -0.421 |
| | (2.797) | (11.351) | (5.192) | (1.645) |
| SOE | -0.065 | 7.742 | -0.014 | -0.288 |
| | (1.172) | (12.717) | (2.018) | (0.386) |
| Competition | 0.771 | 2.106 | 1.267 | 0.508 |
| | (0.948) | (1.859) | (1.663) | (0.535) |
| Year Dummies | YES | YES | YES | YES |
| Firm Dummies | YES | YES | YES | NO |
| Industry Dummies | YES | YES | YES | YES |
| Province Dummies | YES | YES | YES | YES |
| $R^2$ | 0.2602 | 0.4150 | 0.2577 | |

t statistics in parentheses.

* $p<0.1$,

** $p<0.05$,

*** $p<0.01$.

in Table 8. Further, we changed the factors to measure institutional distances. For formal institutional distance, the four factors of political stability, absence of violence, government effectiveness, and regulatory quality were selected. For informal institutional distance, the four indicators of power distance, individualism-collectivism, uncertainty avoidance, and masculinity-femininity were incorporated [51]. As shown in Model (3)-(4) in Table 8, the results remain consistent.

## Discussion and conclusion

In this study, we analyze the data of Chinese enterprises' cross-border M&A events from 2010 to 2017 with a Logit model toward an in-depth investigation of the impact of CSR on the completion of Chinese enterprises' cross-border M&A. This analysis suggests the following conclusions. First, the higher the CSR, the easier it is for Chinese enterprises to complete cross-border M&A. Second, institutional distance moderates the positive effect of CSR on the completion of cross-border M&A. Specifically, formal and informal institutional distance have positive moderating effects. That is, the greater the institutional distance, the greater the positive impact of CSR on the completion of cross-border M&A.

**Table 8. Robustness tests of moderating effects.**

| | (1) | (2) | (3) | (4) |
|---|---|---|---|---|
| | Completion | Completion | Completion | Completion |
| **CSR** | 1.066 | -14.770** | 0.665 | -5.262** |
| | (1.309) | (6.536) | (1.241) | (2.346) |
| **Absolute Formal Distance** | 1.544 | | | |
| | (1.172) | | | |
| **CSR* Absolute Formal Distance** | 3.166** | | | |
| | (1.410) | | | |
| **Absolute Informal Distance** | | 9.848* | | |
| | | (5.326) | | |
| **CSR* Absolute Informal Distance** | | 26.302* | | |
| | | (14.356) | | |
| **4-dimension Formal Distance** | | | 0.440 | |
| | | | (0.510) | |
| **CSR*4-dimension Formal Distance** | | | 1.787* | |
| | | | (0.959) | |
| **4-dimension Informal Distance** | | | | 1.654* |
| | | | | (0.886) |
| **CSR*4-dimension Informal Distance** | | | | 5.861* |
| | | | | (3.525) |
| **Ln_Age** | 18.302 | 13.384 | 14.209 | 51.693** |
| | (15.595) | (15.625) | (15.180) | (24.521) |
| **Ln_Assets** | 5.834** | 2.183 | 5.338** | 6.310 |
| | (2.505) | (4.206) | (2.472) | (4.302) |
| **Cash_ratio** | -1.748 | 15.743 | -1.877 | 1.906 |
| | (2.406) | (11.658) | (2.420) | (5.350) |
| **Flow_ratio** | 2.164 | -5.054 | 2.278 | 0.180 |
| | (1.623) | (3.842) | (1.625) | (1.554) |
| **Asset_liability_ratio** | 10.654 | 16.006 | 13.089* | 14.346 |
| | (8.163) | (27.742) | (7.894) | (21.268) |
| **SOE** | 1.420 | 20.780 | 1.592 | 4.881 |
| | (2.625) | (21.988) | (2.537) | (7.900) |
| **Competition** | 1.267 | 0.693 | 1.215 | 0.883 |
| | (2.865) | (2.317) | (2.807) | (2.631) |
| **Year Dummies** | YES | YES | YES | YES |
| **Firm Dummies** | YES | YES | YES | YES |
| **Industry Dummies** | YES | YES | YES | YES |
| **Province Dummies** | YES | YES | YES | YES |
| $R^2$ | 0.2902 | 0.4334 | 0.2824 | 0.3593 |

As the stakeholder theory argues, successful completion of M&A depends largely on the stakeholders' evaluation of the merged and acquired firm [20]. In cross-border acquisitions, institutional differences between the two firms' home countries accentuate information asymmetries, which may make cooperation and knowledge transfer more problematic, therefore increasing the risks and uncertainties associated with the acquisition [52]. CSR enables merging and acquiring enterprises to gain a positive evaluation among the targeted firms' stakeholders, thereby promoting the completion of M&A, especially when there is a great institutional distance between the host and home countries. As the findings suggest, CSR

could help firms gain market legitimacy in the host countries with formal institutions greatly different from those of China. CSR enables firms to attain more legitimate recognition from stakeholders including employees, governments, the community, and customers when they enter a host country, even if it has different formal institutions.

The research in this study contributes theoretically to the literature on cross-border M&A dynamics by adding to the factors that influence the completion of this process. Through the empirical investigation on the completion of cross-border M&A, we confirm that Chinese enterprises with higher social responsibility are more likely to complete M&As.

This study is among the first to address the institutional context to analyze CSR and illustrate the complexity of the forces underpinning CSR in cross-border M&A. CSR can directly promote the completion of cross-border M&A, though our research suggests that institutional distance influences the impact of CSR on cross-border M&A.

This study also contributes to relevant literature on the influence of institutional distance on M&A by investigating the positive role of CSR in overcoming the negative effect of institutional distances. Different from prior studies solely focusing on the negative influence of institutional distances (e.g., Yang and Boasson [53]), this study further explores how firms actively weaken or remove these negative influences caused by a high level of institutional distance.

This study illustrates the complexities within cross-border M&A and the analytical results will help to improve the probability of cross-border M&A completion. The results provide feasible guidance for the selection of a target destination of cross-border M&A. First, we demonstrate that high CSR is conducive to cross-border M&A. When an enterprise intends to carry out cross-border M&A, highlighting social responsibility will increase the potential to complete the M&A. Second, we propose that the greater the institutional distances between the host and home countries, the greater the impact of CSR on the completion of M&A.

This study has some limitations that also provide promising future research avenues. First, this study investigates the impact of CSR on cross-border M&A in an emerging country using a sample of Chinese enterprises as the merging and acquiring enterprises. Although China provides a relatively emblematic exemplification of emerging countries, an overgeneralization using Chinese data may not comprehensively and specifically reflect the case of cross-border M&A in emerging or developing countries. Thorough research on other emerging countries can yield more universal results. Second, this study focused mainly on the impact of CSR on the completion of M&A under the condition of high institutional distance. In the future, we can further study the impact of CSR on the performance before and after the completion of M&A in the case of high institutional distance.

## Author Contributions

**Conceptualization:** Haiting Li.

**Data curation:** Haiting Li, Xiangcen Zhan.

**Formal analysis:** Shuzhen Li.

**Funding acquisition:** Shuzhen Li.

**Investigation:** Xiangcen Zhan, Mingwei Sun.

**Methodology:** Haiting Li, Shuzhen Li, Xiangcen Zhan, Feng Zhang.

**Project administration:** Shuzhen Li.

**Software:** Xiangcen Zhan.

**Supervision:** Feng Zhang.

**Validation:** Feng Zhang.

**Writing – original draft:** Haiting Li, Shuzhen Li, Mingwei Sun.

**Writing – review & editing:** Haiting Li, Shuzhen Li, Xiangcen Zhan, Feng Zhang.

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
