## [Decision Letter · Decision Letter 0]

25 Jun 2021

PONE-D-21-18638

Corporate Social Responsibility and Cross-border M&A: The Moderating Effect of Institutional Distance

PLOS ONE

Dear Dr. Li,

Thank you for submitting your manuscript to PLOS ONE. After careful consideration, we feel that it has merit but does not fully meet PLOS ONE’s publication criteria as it currently stands. Therefore, we invite you to submit a revised version of the manuscript that addresses the points raised during the review process.

The quantitative framework should be refined. There are more than necessary robustness checks, along with further interpretations.

We look forward to receiving your revised manuscript.

Kind regards,

Stefan Cristian Gherghina, PhD. Habil.

Academic Editor

PLOS ONE

Journal Requirements:

Reviewers' comments:

Reviewer's Responses to Questions

**Comments to the Author**

1. Is the manuscript technically sound, and do the data support the conclusions?

Reviewer #1: Partly

Reviewer #2: Partly

2. Has the statistical analysis been performed appropriately and rigorously? 

Reviewer #1: I Don't Know

Reviewer #2: No

3. Have the authors made all data underlying the findings in their manuscript fully available?

Reviewer #1: No

Reviewer #2: No

4. Is the manuscript presented in an intelligible fashion and written in standard English?

Reviewer #1: Yes

Reviewer #2: Yes

5. Review Comments to the Author

Reviewer #1: I would like to thank the authors and the editor to have to opportunity to review this paper.

The subject is interesting, and the paper is overall clear and well written. I hope my comment will enable the authors to improve their work.

The study is performed concerning Chinese firms, which is interesting, but the reader has almost no information on the specificities of the Chinese context concerning M&A

The way hypothesis are presented is strange. There are two competing hypothesis at the beginning (H1a and H1b) but afterwards, the authors take a positive relationship for granted for H2 and H3.

You are focusing on the CSR on the targeting firm. Why not about the CSR of the target firm?

The stakeholder vs shareholder perspective is sometimes difficult to articulate and I think you should explain more in depth the moderating relationship: why should institutional distance moderate the precise CSR – M&A completion relationship? This is for me not very clear in the paper.

I’m not specifically familiar about formal and informal institutions but the moderating effect is strange. The two concepts should be close (and are significantly correlated, see the correlation table). But the results are opposite. My major concern is here. I fear that the model is over identified since you add twice almost the same variable. Even if your VIF are under 10, it has an impact on your coefficients. You should include one variable or the other (formal or informal) but not both.

Besides, the results are then difficult to interpret. The two institutional distances are very close together but give totally opposite results, which is difficult to understand and I am not convinced that it is not just a spurious result. When using CSR2, the variable becomes unsignificant.

I’m wondering about other variables that should be introduced in the model: valuation of the target firm (market to book), if it is an hostile transaction or growth of the target.

Endogeneity problems may arise. How did you deal with it? Would PSM or diff in diff be interesting in your case?

I’m not familiar with Hexun.com but some studies concerning China are using RKS, which could offer another interesting proxy for CSR

A few minor typos or awkward sentences :

(1.) “less gratifying”, “unavoidably”

(2.2) “There still a gap”

(3.2.2) “institutional diatance”

Reviewer #2: Corporate Social Responsibility and Cross-border M&A: The Moderating Effect

of Institutional Distance

This paper investigates the impact of corporate social responsibility (CSR) on the completion of cross-border M&A with the moderating role of institutional distance over the years 2010-2016. The results suggest CSR has a positive role in the completion of the cross-border M&A of Chinese enterprises. Second, the formal institutional distance moderates the effect of CSR positively, whereas informal institutional distance moderates the effect negatively.

Comments:

In my view, this is a potentially interesting paper. However, I have the following concerns with this study.

The paper should provide a table of the sample selection and exclusions. How is it distributed in time? Year-wise classifications between completed and non-completed cross-border M&A deals would add a clearer view of the sample. It should be better to have a distribution of target countries. I would also suggest you should consider providing descriptive statistics of all variables used in this study between the completed and non-completed M&A deals and test between these two groups using parametric and non-parametric tests to see if there are any significant differences between the above two groups.

The biggest concern in the analysis is correlated omitted variables. There could be unobserved factors that are correlated with CSR, institutional distance and control variables. This raises concerns regarding omitted variable bias in your probit regression results. My suggestion is to run an instrumental variable (IV) probit model to address the potential endogeneity concern.

This study examines the impact of CSR on the completion of cross-border M&A using a logit model. The dependent variable (Completion) is a binary variable that whether the cross-border M&A was completed or not. I would suggest that you should calculate short-run abnormal returns and long-run performance around the cross-border M&A deals and test the moderating effect of CSR and institutional distance on the calculated performance.

6. PLOS authors have the option to publish the peer review history of their article (what does this mean?). If published, this will include your full peer review and any attached files.

Reviewer #1: No

Reviewer #2: No

---

## [Author Response · Author response to Decision Letter 0]

3 Sep 2021

Response to the Comments of Academic Editor

Comments: Thank you for submitting your manuscript to PLOS ONE. After careful consideration, we feel that it has merit but does not fully meet PLOS ONE’s publication criteria as it currently stands. Therefore, we invite you to submit a revised version of the manuscript that addresses the points raised during the review process. 

The quantitative framework should be refined. There are more than necessary robustness checks, along with further interpretations. 

Response: Thank you very much for your positive remarks on our manuscript. We greatly appreciate for allowing us to revise and improve our manuscript. Following the comments of you, Reviewer 1 and Reviewer 2, we have carefully refined the quantitative framework and clearly clarified the statements. We hope you and the reviewers will find our response and revision work satisfactory. 

Please see our responses to Reviewer 1 and Reviewer 2 below.

 

Response to the Comments of Reviewer #1

The subject is interesting, and the paper is overall clear and well written. I hope my comment will enable the authors to improve their work.

Response: Thank you very much for your positive remarks on our paper. Following your guide, we have carefully revised the paper. We believe that your comments have improved the quality of this paper significantly.

Comments: The study is performed concerning Chinese firms, which is interesting, but the reader has almost no information on the specificities of the Chinese context concerning M&A.

Response: Thank you very much for your kind and constructive suggestions. Following your suggestions, we have addressed your concern on the information about the specificities of the Chinese M&A by providing relevant data in our sample years 2010-2016. This point has been clarified in our manuscript (pp.3-4):

Specifically, this study utilizes a sample of Chinese firms as the research context, for two reasons. First, according to the statistics from open sources, the money spent on cross-border M&A transactions by Chinese companies greatly increased, from 34.7 billion dollars in 2010 to 150.6 billion dollars in 2016, and the numbers of transactions climbed from 268 to 738. Furthermore, with the reinforcement of China’s “Go Global” strategy and the Belt and Road initiative, as embodied in the upgrading of corporations, the process of cross-border M&A by Chinese enterprises is growing significantly. For example, Chinese companies investing in these countries conducted no less than 192 M&A transactions in 2016, representing an aggregate value of 19.3 billion dollars, in contrast with 46 transactions and 4.6 billion dollars in 2010. However, the completion rate of cross-border M&A by Chinese enterprises seems below the world average. This outcome may indicate Chinese enterprises may not notice the underlying factors that may lead to failure in their cross-border M&A activities. 

Second, China, the largest emerging market with the transition into the market economy, is integrating into the world economy and enhancing the governance quality, thereby attracting more and more academic attention. Considering that there are many common characteristics between China and other emerging markets, this study advances our knowledge of emerging markets. Additionally, it may contribute the stream of literature on M&As through investigating how firms from emerging markets complete cross-border M&A deals, especially considering firms from these regions experience high percentage collapse before completion (Zhou et al., 2016).

Comments: The way hypothesis are presented is strange. There are two competing hypothesis at the beginning (H1a and H1b) but afterwards, the authors take a positive relationship for granted for H2 and H3.

Response: Thank you very much for your constructive comments. As you suggested, we have refined the hypothesis to more reflect our findings by drawing upon extent studies and focusing on the characteristics of emerging markets. We argue that the positive relationship between CSR of Chinese companies and M&A is more common. Namely, CSR has a significant positive effect on cross-border M&A. That is, the stronger the CSR, the more likely an enterprise will be to complete a cross-border M&A. This part has been clarified in our manuscript (pp.6-7). 

Specifically, we suppose that CSR has a positive effect on the completion of M&A for Chinese companies. As characterized by the emerging market, institutions are transitioning to the market economy, their implementation can still be erratic, and their norms are evolving (Bruton et al., 2019). Consequently, firms from emerging markets like China may be likely to suffer the liability of foreignness when conducting cross-border M&A in international markets. Accordingly, as a strategic tactic, CSR enables these firms to enhance their legitimacy in host countries and build trustworthiness with target firms’ stakeholders (Bertrand et al., 2021), thereby overcoming the liability of foreignness (Husted et al., 2016) and facilitating the completion of M&A. Therefore, we suppose that,

Hypothesis 1: CSR has a significant positive effect on cross-border M&A. That is, the stronger the CSR, the more likely an enterprise will be to complete a cross-border M&A.

Comments: You are focusing on the CSR on the targeting firm. Why not about the CSR of the target firm?

Response: Thank you very much for your constructive comments. As you suggested, we definitely believe that the effect of CSR of the target firm on M&A would be a very interesting topic, so it will be investigated further in our future research. Specifically, current study emphasizes the CSR on the targeting firm rather than the target firm for three reasons. 

Firstly, the antecedents on the completion of M&As have been attracting the scholarly attentions. This study aims to explore how firms from emerging market enhance the likelihood of M&A in international markets. To achieve this research objective, we focus on the CSR of the targeting firm and investigate how firms from China overcome the liability of foreignness through increasing their CSR performance. As stated in this study, our findings provide implications for firms from emerging markets.

Secondly, as sated by the prior studies, a variety of reasons from the targeting firm may cause the failure of a deal such as shareholder opposition, financing problems or internal target resistance (Arouri et al., 2019). In other words, the outcome of M&A deals depends on the assessment and opinion of many of the targeting firm’s stakeholders (Arouri et al., 2019). Therefore, it is meaningful and important to discuss the role of targeting firm’s CSR.

Finally, compared to the target firm, it is more important and costly for the targeting firm to complete the cross-border M&A due to upfront financial costs and termination fees as well as losses in terms of firm reputation, credibility, time and diversion of managerial attention (Hawn, 2014). 

Comments: The stakeholder vs shareholder perspective is sometimes difficult to articulate and I think you should explain more in depth the moderating relationship: why should institutional distance moderate the precise CSR – M&A completion relationship? This is for me not very clear in the paper.

Response: Thank you very much for your kind and constructive comments. Following your guide, we further clarified the moderating relationship and strengthen the arguments (pp.7-8).

 The main logic is that institutional distance raises the risk and uncertainty of firms’ internationalization and M&A (Kostova et al., 2020), while CSR may reduce the uncertainties (Arouri et al., 2019). Specifically, we develop our arguments from three points (pp.7-8).

Firms looking to invest overseas face a natural disadvantage when they conduct multinational operations as “outsiders”, especially when institutional distance is high. Enterprises entering a host country’s market face mainly the problem of local unfamiliarity caused by information asymmetry, along with discrimination due to the lack of legitimacy and relationship harm as they lack embeddedness. First, the unfamiliarity hazard mainly refers to the fact that overseas enterprises entering a host country’s market are at a disadvantage in understanding the host country’s market situation, legal system, and culture. Obtaining such information often requires a relatively high cost and high engagement in CSR to gain trust and recognition among the host countries’ stakeholders. For example, Kostova and Zaheer (2020) find that when the host and home countries have high institutional distance, CSR is more attractive for foreign firms, and the more positive its CSR activities are, the easier it is for them to correctly understand the host country’s institutional environment, especially informal implicit social norms, cultural practices, religious beliefs, and conventional rules. 

Second, the harm caused by the lack of legitimacy refers to the lack of information to judge a foreign firm’s operations in the host country, where locals often rely on stereotypes to judge the behavior of the company. If the foreign firm has higher engagement in positive CSR activities or show successful investment behaviors, then the host country’s stakeholders will believe that the most obvious characteristics are positive and reliable. 

Finally, the damage to relationships caused by the lack of embeddedness mainly refers to the fact that a new foreign firm does not effective and close relationships with the host country government, suppliers, consumers, communities, and other stakeholders. Thus, firms establish relations with these parties by engaging in various CSR activities to achieve cross-border M&A. Therefore, we examine the moderating effect of formal and informal institutional distance individually.

Comments: I’m not specifically familiar about formal and informal institutions but the moderating effect is strange. The two concepts should be close (and are significantly correlated, see the correlation table). But the results are opposite. My major concern is here. I fear that the model is over identified since you add twice almost the same variable. Even if your VIF are under 10, it has an impact on your coefficients. You should include one variable or the other (formal or informal) but not both.

Response: Thank you very much for your insightful comments. We appreciate your kind guidance that helped us significantly improved the quality of our paper. Taking into consideration your concern of the high correlation, we adjust the measurements of formal institutions and informal institutions and then run the regressions of both models containing each interaction term respectively. After adjustment, the coefficient of correlation is lowed to 0.0921 and the empirical results are shown below (pp.18-20).

As reported in Model (3) of Table 6, the interaction term between formal institutional distance and CSR is significantly positive at the 10% confidence level. That is, in the presence of formal institutional distance, companies with high CSR have stronger incentives to conduct M&A activities, supporting Hypothesis 2. Model (4) of Table 6 examines the moderating effect of informal institutional distance, and the interaction term is significantly negative at the 10% confidence level. In other words, the greater informal institutional distance is, the less positive is the impact of CSR on the completion of cross-border M&A. Hence, informal institutional distance negatively moderates the positive impact of CSR on the completion of cross-border M&A, which is contrary to Hypothesis 3.

Table 6. Basic Results and Moderating Effects.

Comments: Besides, the results are then difficult to interpret. The two institutional distances are very close together but give totally opposite results, which is difficult to understand and I am not convinced that it is not just a spurious result. When using CSR2, the variable becomes unsignificant.

I’m wondering about other variables that should be introduced in the model: valuation of the target firm (market to book), if it is a hostile transaction or growth of the target. 

Response: Thank you very much for your detailed comments. Carefully following your suggestions, we have revised the measurements of the two institutional distances in order to lower correlation coefficients and reported the results shown above. What’s more, we also believe that if valuation of the target firm was introduced in the model would help us get interesting conclusions. However, limited by the availability of data, we cannot get any result as financial indicators of the target firm due to a large amount of missing data. Making full use of existing data, the OECD dummy variable measuring whether the target firm is from OECD countries is included in our model. 

Comments: Endogeneity problems may arise. How did you deal with it? Would PSM or diff in diff be interesting in your case? 

Response: Thank you very much for your kind and constructive comments. 

Although we did some robustness tests including changing the econometric method and constructing new measurements of CSR, there are still potential endogeneity problems resulting from omitted variables. To ensure unbiased estimators, we use the Propensity Score Matching (PSM) method to run the regression (pp.22-24). Specifically, we divide the sample into two groups based on industry-mean CSR: a treatment group where companies have higher CSR scores than the mean in their respective industries and a comparison group with lower CSR. Then, we apply 1-to-1 nearest neighbor matching method within caliper to balance the observable individual characteristics between these two groups according to variables that influence the possibility of entering the treatment group. After matching, we estimate the model using the matched sample. The result in column (1) in Table 8 provides evidence consistent with the main conclusion, thus supporting our main analysis. 

Moreover, we used an instrumental variable (IV) probit analysis. Drawing upon prior studies (El Ghoul et al. 2011; Harjoto and Jo 2015; Benlemlih and Bitar 2018; Ozdemir et al., 2021), we use industry-mean CSR as the instrumental variable for CSR. Industry-mean CSR is correlated to firm-level CSR as the intensity of CSR from peer companies in the same industry motivates companies to engage more in CSR activities, and it is not linked to cross-border M&A for a particular target (Ozdemir et al., 2021). The first stage regresses CSR on the instrumental variable (industry-mean CSR) with all other exogenous variables. Then, we regress cross-border M&A on the predicted CSR from the first stage. The results are reported in columns (2) and (3) in Table 8, respectively. The IV results are consistent with the basic findings, showing the robustness of our results.

Comments: I’m not familiar with Hexun.com but some studies concerning China are using RKS, which could offer another interesting proxy for CSR. 

Response: Thank you very much for your detailed comments. Mentioned in our original manuscript, we use CSR data from Hexun.com as it built the first vertical financial portal website in China and developed as an authorized financial securities information provider, as well as providing CSR score data for all listed companies according to CSR reports released on the official websites or in the annual reports issued by listed companies, thus providing us with accurate results. Following your suggestions, we also matched the CSR indicator from the RKS with our cross-border M&As data. However, the size of matched samples is just over 100 and there are many missing values, so we cannot run the regression using this small sample in order to get unbiased conclusion. In the future, we will continually investigate this study upon the availability of necessary data. 

Comments: A few minor typos or awkward sentences:

(1.) “less gratifying”, “unavoidably”

(2.2) “There still a gap”

(3.2.2) “institutional diatance”

Response: Thank you very much for your detailed comments. Following your suggestions, we have revised these typos and awkward sentences with the help of professional copyeditor. Much more than this, we checked the whole manuscript supplemented with the latest revisions in carefulness and responsible attitude. 

Thank you very much again for all your patience and insightful comments and suggestions. Hopefully, our response and revision could address your concerns. In any case, we believe that the quality of our paper has been improved with your kind help in this review process.

 

Response to the Comments of Reviewer #2

This paper investigates the impact of corporate social responsibility (CSR) on the completion of cross-border M&A with the moderating role of institutional distance over the years 2010-2016. The results suggest CSR has a positive role in the completion of the cross-border M&A of Chinese enterprises. Second, the formal institutional distance moderates the effect of CSR positively, whereas informal institutional distance moderates the effect negatively. In my view, this is a potentially interesting paper. However, I have the following concerns with this study.

Response: Thank you very much for your positive remarks on our paper. Very carefully following your guide, we have revised the paper, especially the part of methodology. Your comments have improved the quality of our paper significantly. Hopefully, our revisions could be satisfactory to you. 

Comments: The paper should provide a table of the sample selection and exclusions. How is it distributed in time? Year-wise classifications between completed and non-completed cross-border M&A deals would add a clearer view of the sample. It should be better to have a distribution of target countries. I would also suggest you should consider providing descriptive statistics of all variables used in this study between the completed and non-completed M&A deals and test between these two groups using parametric and non-parametric tests to see if there are any significant differences between the above two groups.

Response: Thank you very much for your insightful comments. Following your suggestions, we present year-wise classifications between completed and non-completed cross-border M&A deals, a distribution of target countries between these two groups, summary statistics of all variables classified according to the sample mean and median of the CSR scores as well as testing the significance of differences. These results are provided in Table 2, Table 3 and Table 4 respectively (pp.13-15). As shown in Table 3, firms completing cross-border M&A deals have higher CSR scores than firms with failure of overseas M&A, suggesting that firm with higher CSR are more possible to succeed in cross-border M&A deals. 

Table 2. Sample Distribution by Year.

Table 4. Summary Statistics by Completion.

Comments: The biggest concern in the analysis is correlated omitted variables. There could be unobserved factors that are correlated with CSR, institutional distance and control variables. This raises concerns regarding omitted variable bias in your probit regression results. My suggestion is to run an instrumental variable (IV) probit model to address the potential endogeneity concern.

Response: Thank you very much for your constructive suggestions. 

To address concerns of potential endogeneity problem arising from omitted variables, we have done the instrumental variable (IV) probit analysis as you suggested (pp. 23-24). Drawing upon prior studies (El Ghoul et al. 2011; Harjoto and Jo 2015; Benlemlih and Bitar 2018; Ozdemir et al., 2021), we use industry-mean CSR as the instrumental variable for CSR. Industry-mean CSR is correlated to firm-level CSR as the intensity of CSR from peer companies in the same industry motivates companies to engage more in CSR activities, and it is not linked to cross-border M&A for a particular target (Ozdemir et al., 2021). The first stage regresses CSR on the instrumental variable (industry-mean CSR) with all other exogenous variables. Then, we regress cross-border M&A on the predicted CSR from the first stage. The results are reported in columns (2) and (3) in Table 8, respectively. The IV results are consistent with the basic findings, showing the robustness of our results.

Moreover, we use Propensity Score Matching (PSM) method to run the regression following your suggestions (pp.22-24). Specifically, we divide the sample into two groups based on industry-mean CSR: a treatment group where companies have higher CSR scores than the mean in their respective industries and a comparison group with lower CSR. Then, we apply 1-to-1 nearest neighbor matching method within caliper to balance the observable individual characteristics between these two groups according to variables that influence the possibility of entering the treatment group. After matching, we estimate the model using the matched sample. The result in column (1) in Table 8 provides evidence consistent with the main conclusion, thus supporting our main analysis. 

Table 8. PSM and IV Tests.

Comments: This study examines the impact of CSR on the completion of cross-border M&A using a logit model. The dependent variable (Completion) is a binary variable that whether the cross-border M&A was completed or not. I would suggest that you should calculate short-run abnormal returns and long-run performance around the cross-border M&A deals and test the moderating effect of CSR and institutional distance on the calculated performance.

Response: Thank you very much for your constructive comments. Your comments provide us another insightful implication to move our research forward in the future. 

First, the completion of M&A has been attracting the academic attentions, especially involving emerging market (e.g., Zhou, Xie, and Wang, 2016; Dikova, Sahib, and Witteloostuijn, 2010). Additionally, there is a high percentage collapse before completion (Zhou, Xie, and Wang, 2016). Therefore, it is meaningful to discuss how to enhance the success of M&A.

Second, we definitely believe that it is a very interesting topic to discuss the influence of CSR and institutional distance on the abnormal returns caused by M&A deals. However, we suppose this is another topic that is different from the current topic exploring the influence of CSR and institutional distance on the completion of M&A. Furthermore, as you suggested, it seems that CSR may has an indirect effect (e.g., the moderating effect) on the abnormal returns caused by M&A, as the direct link between CSR and the abnormal returns caused by M&A is not very clear. Thus, we need to develop very strong arguments different with current study to answer why and how CSR affects the abnormal returns closely related to the event of M&A. After careful consideration, we suppose that it is better to regard it as quite another research rather than integrating it into the current topic. We appreciate your inspiring comments leading us to conduct potential research in the future.

Thank you very much again for all your patience and insightful comments and suggestions. Hopefully, our response and revision could address your concerns. In any case, we believe that the quality of our paper has been improved with your kind help in this review process.

---

## [Decision Letter · Decision Letter 1]

28 Sep 2021

PONE-D-21-18638R1Corporate Social Responsibility and Cross-border M&A: The Moderating Effect of Institutional DistancePLOS ONE

Dear Dr. Li,

Thank you for submitting your manuscript to PLOS ONE. After careful consideration, we feel that it has merit but does not fully meet PLOS ONE’s publication criteria as it currently stands. Therefore, we invite you to submit a revised version of the manuscript that addresses the points raised during the review process.The paper still requires further revisions with reference to empirical design, discussion, as well as paper contribution.

We look forward to receiving your revised manuscript.

Kind regards,

Stefan Cristian Gherghina, PhD. Habil.

Academic Editor

PLOS ONE

Reviewers' comments:

Reviewer's Responses to Questions

**Comments to the Author**

1. If the authors have adequately addressed your comments raised in a previous round of review and you feel that this manuscript is now acceptable for publication, you may indicate that here to bypass the “Comments to the Author” section, enter your conflict of interest statement in the “Confidential to Editor” section, and submit your "Accept" recommendation.

Reviewer #1: (No Response)

2. Is the manuscript technically sound, and do the data support the conclusions?

Reviewer #1: Partly

3. Has the statistical analysis been performed appropriately and rigorously? 

Reviewer #1: I Don't Know

4. Have the authors made all data underlying the findings in their manuscript fully available?

Reviewer #1: Yes

5. Is the manuscript presented in an intelligible fashion and written in standard English?

Reviewer #1: Yes

6. Review Comments to the Author

Reviewer #1: Dear authors,

Thanks for your work on your research. I think the Chinese context is now well explained, the hypothesis and the paper more clear overall.

I, however, still have a few comments:

Measures

The way you measure formal and informal distance is not clear. For the FID, you use absolute values whereas, most of the time, Euclidean distance is preferred (see Wanli Li, Chaohui Wang, Qizhe Ren, Ding Zhao, 2020, Institutional distance and cross-border M&A performance: A dynamic perspective, Journal of International Financial Markets, Institutions and Money, Volume 66). You mention Fi but they are not really defined: is it factors from the WBI? Concerning informal distance, I do not understand why you only use one dimension of Hofstede. Keig and al [46] use four of the dimensions, not one.

Specification

I’m still confused by your specification since there are important missing variables (like valuation, cash, firm growth if there is a runup, the age of the firm…) and in the same time, most of the variables (as size) do not display a significant relationship. How could we convince us that the model is correctly specified?

Robustness

Concerning the robustness tests, they are performed on H1, but not on what is the most interesting in the paper, H2 and H3. This is an important issue for the robustness of your findings. I’m also not really convinced by your explanation of the counterintuitive results of H3. This reference may help you: Javier Aguilera-Caracuel, Nuria Esther Hurtado-Torres, Juan Alberto Aragón-Correa, Alan M. Rugman, 2013, Differentiated effects of formal and informal institutional distance between countries on the environmental performance of multinational enterprises, Journal of Business Research, Volume 66, Issue 12.

Contribution

Besides, you should explain your contribution compared to: Yang, K., & Boasson, V. (2021). Cross-border acquisitions and institutional distance: does country connectedness matter? International Journal of Business and Globalisation, 28(3), 332-348.

I hope these comments will improve your work.

Best regards,

7. PLOS authors have the option to publish the peer review history of their article (what does this mean?). If published, this will include your full peer review and any attached files.

Reviewer #1: No

---

## [Author Response · Author response to Decision Letter 1]

11 Nov 2021

Response to the Comments of Academic Editor

Comments: Thank you for submitting your manuscript to PLOS ONE. After careful consideration, we feel that it has merit but does not fully meet PLOS ONE’s publication criteria as it currently stands. Therefore, we invite you to submit a revised version of the manuscript that addresses the points raised during the review process.

The paper still requires further revisions with reference to empirical design, discussion, as well as paper contribution. 

Response: Thank you very much for your positive remarks on our manuscript. We very much appreciate you allowing us to revise and improve the paper again. Following the comments of you and Reviewer 1, we have carefully revised the empirical design, clearly clarify our contribution and discussion. We hope you and the reviewer will find our response and revised manuscript satisfactory.

Response to the Comments of Reviewer #1

Comments: Dear authors,

Thanks for your work on your research. I think the Chinese context is now well explained, the hypothesis and the paper more clear overall.

I, however, still have a few comments:

Response: Thank you very much for your positive remarks on our paper. Following your suggestions, we have carefully revised the paper again. We are very sure that your remarks have improved the quality of this paper greatly. Specifically, your constructive suggestions have helped us to resolve the major concerns from measures and model specifications and make the contributions of our paper much clearer and accurate.

Comments: Measures

The way you measure formal and informal distance is not clear. For the FID, you use absolute values whereas, most of the time, Euclidean distance is preferred (see Wanli Li, Chaohui Wang, Qizhe Ren, Ding Zhao, 2020, Institutional distance and cross-border M&A performance: A dynamic perspective, Journal of International Financial Markets, Institutions and Money, Volume 66). You mention Fi but they are not really defined: is it factors from the WBI? Concerning informal distance, I do not understand why you only use one dimension of Hofstede. Keig and al [46] use four of the dimensions, not one. 

Response: Thank you very much for your kind and constructive suggestions. Carefully following your suggestions, we have revised the measurements and calculations of institutional distance. Now, we believe that the measurements are more reliable and accurate.

In reference to Li et al. (2020) as you suggested, we have re-calculated both formal and informal institutional distance by using the Euclidean distance. 

In this study, we use the Worldwide Governance Indicators (WGI) to measure formal institutions. The WGI project measures the development of a country’s institutions by using six indicators: (1) Voice and Accountability; (2) Political Stability and Absence of Violence; (3) Government Effectiveness; (4) Regulatory Quality; (5) Rule of Law; and (6) Control of Corruption. Specifically, 〖FID〗_j represents the formal institutional distance between the host country j and the home country, I_nj is the score of dimension n for the host country in year t, I_nc denotes the score of dimension n for the home country in the year t, and V_n denotes the variance of dimension n in the same year. 

〖FID〗_j=√(∑_n^6▒〖(I_nj-I_nc)〗^2/V_n )

In the prior version, we argue that uncertainty avoidance, one of Hofstede’s cultural dimensions, is more closely related with the perception and acceptance of MNCs’ CSR practices from stakeholders in host country, thereby selecting it to measure informal institutions. 

In the new revision, following your suggestions, we measured informal institutions by using the whole six dimensions of Geert Hofstede's national culture: (1) Individualism; (2) Power Distance; (3) Masculinity; (4) Uncertainty Avoidance; (5) Long-term Orientation; and (6) Indulgence. And also, we re-calculated the informal institutional distance by using the Euclidean distance. 

〖IID〗_j=√(∑_m^6▒〖(I_mj-I_mc)〗^2/V_m )

Comments: Specification

I’m still confused by your specification since there are important missing variables (like valuation, cash, firm growth if there is a runup, the age of the firm…) and in the same time, most of the variables (as size) do not display a significant relationship. How could we convince us that the model is correctly specified?

Response: Thank you very much for your detailed and constructive comments about the model specification. We very much agree with you that the possible omission of some important variables may influence the accuracy of our regression estimations. To resolve the concerns about the misspecification of model, especially the omission of some important variables, we made a major revision on the model specification.

First, besides of year, industry, and region fixed effect, we added firm fixed effect as well, so that we could rigorously control for all other non-observational firm-level variables in the model, thereby ensuring the estimation accuracy.

Second, at the merging and acquiring enterprise level, this paper controls for the following variables: (1) Firm age (Ln_Age) is the natural logarithm of the number of years since the merging and acquiring company started operating. (2) Firm size (Ln_Assets) is the natural logarithm of the total assets of the merging and acquiring company at the end of the year. (3) Cash flow (Cash_ratio) is the ratio of the sum of cash and cash equivalent to current liabilities, which reflects the operation of enterprise. (4) Working capital ratio (Flow_ratio) is the ratio of current assets to current liabilities measuring the risk of operation, which is vital for cross-border M&A. (5) Asset_liability_ratio is the ratio of total liabilities to total assets, which is a measurement of financial leverage. (6) Equity ownership (SOE), which is a binary variable equal to 1 if the acquiring company is a state-owned enterprise, and 0 otherwise. State ownership ties are likely to influence the regulated resources and policy treatment that a firm can acquire from the government, which may affect the competition of an M&A event. (7) Competition is measured by the number of competing buyers for overseas M&A.

Further, the IV estimation method could also reduce our concern about model misspecification caused by the missing variables.

Table 1 reports the estimated Logit model results based on the revised model specification. Model (1) is the baseline regression. Model (2) reports the regression results with control variables. In Model (2), the regression coefficients of controls such as Ln_Assets, Flow_ratio and Asset_liability_ratio are significantly positive, denoting that the good financial performances of the acquiring enterprises could motivate overseas M&A. The regression coefficient of CSR is significantly positive, indicating that CSR has a significant positive relationship with Chinese enterprises’ cross-border M&A. Model (3) and model (4) show the moderating effects of formal institutional distance and informal institutional distance respectively after controlling all fixed effects and variables.

Specifically, upon carefully revised the measurement and model specification, we finally found a positive moderating effect of informal institutional distance, in opposite to our prior findings. To re-confirm the consistence of the moderating effect, we conducted a series of robustness check for it and still found the same result. Therefore, we very much believe that the significant improvement of the measurement and model specification lead to the “opposite” but “appropriate” regression estimation. 

We appreciate very much for your great help! Your insightful suggestions help us to correct the model specification and variable measurements.

Table 1 Basic Results and Moderating Effects

Comments: Robustness

Concerning the robustness tests, they are performed on H1, but not on what is the most interesting in the paper, H2 and H3. This is an important issue for the robustness of your findings. I’m also not really convinced by your explanation of the counterintuitive results of H3. This reference may help you: Javier Aguilera-Caracuel, Nuria Esther Hurtado-Torres, Juan Alberto Aragón-Correa, Alan M. Rugman, 2013, Differentiated effects of formal and informal institutional distance between countries on the environmental performance of multinational enterprises, Journal of Business Research, Volume 66, Issue 12.

Response: Thank you very much for your insightful comments. Following your suggestions, we further conducted the robustness tests for the moderating effects through changing the measurement and calculation of institutional distance, and we still found the consistent results. For example, we used the absolute value to measure institutional distance, as displayed in Model (1)-(2) in Table 2. And also, we changed the factors to measure institutional distances. For formal institutional distance, the four factors of political stability, absence of violence, government effectiveness, and regulatory quality were selected. For informal institutional distance, the four indicators of power distance, individualism-collectivism, uncertainty avoidance, and masculinity-femininity were selected.

Thank you very much again for your insightful comments, especially your persistent focus on the “counterintuitive” moderating effect of informal institutional distance and your great help on model specification and variable measurement.

Table 2 Robustness Tests of Moderating Effects

Comments: Contribution

Besides, you should explain your contribution compared to: Yang, K., & Boasson, V. (2021). Cross-border acquisitions and institutional distance: does country connectedness matter? International Journal of Business and Globalisation, 28(3), 332-348.

I hope these comments will improve your work.

Response: Thank you very much for your kind and constructive comments. Definitely, your guide has greatly helped us to improve the quality of our paper, make our results more reliable, and refine the theoretical contributions. We appreciate your suggestions and help to our manuscript greatly.

Specifically, we carefully read the reference of Yang and Boasson (2021) you suggested and reconsider our “uniqueness” from Yang and Boasson (2021), and then clarify the theoretical contributions again.

First, the research focuses of our study significantly differ from Yang and Boasson (2021). Yang and Boasson (2021) focuses on the main effect of institutional distance on cross-border M&As, especially the interaction of formal and informal institutional distance, whilst this study focuses on the vital role of corporate social responsibility in cross-border M&As, especially its positive effect on overcoming the barriers of institutional distance.

Second, this study contributes to the literature relevant with the influence of institutional distance on M&A (e.g., Yang and Boasson, 2021) through investigating the positive role of CSR on overcoming the negative effect of institutional distances. That is, different from the prior studies just focusing on the negative influence of institutional distances, this study further explores how firms actively do to weaken or remove these negative influences caused by a high level of institutional distance.

Thank you very much again! We appreciate for your great help to correct our model and improve the quality of our paper. Hopefully, our revisions could make you satisfactory. 

References:

 Li WL, Wang CH, Ren QZ, Zhao D. Institutional distance and cross-border M&A performance: A dynamic perspective. Journal of International Financial Markets, Institutions and Money. 2020;66. doi: 10.1016/j.intfin.2020.101207.

 Kisgen DJ, Qian J, Song W. Are fairness opinions fair? The case of mergers and acquisitions. J Financ Econ. 2009;91(2):179-207. doi: 10.1016/j.jfineco.2008.03.001.

 Chen Y, Huang Y, Chen C. Financing constraints, ownership control, and cross-border M&As: evidence from nine East Asian economies. Corp Gov Int Rev. 2009;17: 665-680. doi: 10.1111/j.1467-8683.2009.00770.x. 

 Xia J, Ma X, Lu JW, Yiu DW. Outward foreign direct investment by emerging market firms: a resource dependence logic. Strateg Manag J. 2014;35(9):1343-1363. doi: 10.1002/smj.2157.

 Tang Q, Gu FF, Xie E, Wu Z. Exploratory and exploitative OFDI from emerging markets: impacts on firm performance. Int Bus Rev. 2020; 29:101661. doi: 10.1016/j.ibusrev.2019.101661.

 Keig DL, Brouthers LE, Marshall VB. The impact of formal and informal institutional distances on MNE corporate social performance. Int Bus Rev. 2019;28:85-92. doi: 10.1016/j.ibusrev.2019.05.004.

---

## [Decision Letter · Decision Letter 2]

9 Dec 2021

PONE-D-21-18638R2Corporate Social Responsibility and Cross-border M&A: The Moderating Effect of Institutional DistancePLOS ONE

Dear Dr. Li,

Thank you for submitting your manuscript to PLOS ONE. After careful consideration, we feel that it has merit but does not fully meet PLOS ONE’s publication criteria as it currently stands. Therefore, we invite you to submit a revised version of the manuscript that addresses the points raised during the review process.

The revised version of the manuscript improved in a proper manner. However, there are still required revisions with reference to hypotheses formulation, quantitative framework, as well as concluding remarks. Not least, the English language should be thoroughly improved.

We look forward to receiving your revised manuscript.

Kind regards,

Stefan Cristian Gherghina, PhD. Habil.

Academic Editor

PLOS ONE

Journal Requirements:

Reviewers' comments:

Reviewer's Responses to Questions

**Comments to the Author**

1. If the authors have adequately addressed your comments raised in a previous round of review and you feel that this manuscript is now acceptable for publication, you may indicate that here to bypass the “Comments to the Author” section, enter your conflict of interest statement in the “Confidential to Editor” section, and submit your "Accept" recommendation.

Reviewer #3: All comments have been addressed

2. Is the manuscript technically sound, and do the data support the conclusions?

Reviewer #3: Yes

3. Has the statistical analysis been performed appropriately and rigorously? 

Reviewer #3: Yes

4. Have the authors made all data underlying the findings in their manuscript fully available?

Reviewer #3: Yes

5. Is the manuscript presented in an intelligible fashion and written in standard English?

Reviewer #3: Yes

6. Review Comments to the Author

Reviewer #3: 1. The authors have followed the previous comments from the editor and reviewers to make revisions in accordance.

2. The statements in hypotheses 1 to 3 are not consistent with the usual academic writing. In a hypothesis, there will not be the term ‘significantly’. Significance (or insignificance) is the result from statistical tests. Therefore, Hypothesis 1 may be like “CSR has a positive effect on cross-border M&A.” Usually there will not be a sentence of “That is,…” in a research hypothesis. Research hypotheses 2 and 3 are also strange because they are not expressed in complete sentences. They may be revised as “M positively (or negatively) moderates the effect of X on Y.”

3. However, the dataset are during 2010-2016. Tables 6-7 are results from fixed-effects panel regressions. Since each M&A is a one-shot event, maybe it is fine to use fixed-effects regression instead of random-effects data regression. The authors need to justify why a fixed-effects Logit regression is appropriate to use for the data analysis. If possible, some statistical tests should be provided to justify the use of fixed-effects regressions.

4. The current conclusion now can be directly supported from the empirical findings in this paper. The authors’ efforts in revising this paper should be positively affirmed.

5. There are still some typos and format problems in the revised manuscript which need to be carefully corrected. For instance, in the Figure, H1, H2, and H3 cover parts of the arrows (paths). On page 38, it shows “42. 42.”

7. PLOS authors have the option to publish the peer review history of their article (what does this mean?). If published, this will include your full peer review and any attached files.

Reviewer #3: No

---

## [Author Response · Author response to Decision Letter 2]

20 Dec 2021

PONE-D-21-18638

Response: Thank you very much for your positive remarks on our manuscript. We arr thankful for allowing us to revise and improve the paper again. Following the comments of you and Reviewer 3, we have carefully revised the expression of hypothesis, clarified our empirical model and corrected typos. We hope you and the reviewer will find our responses and revision work satisfactory. 

Response to the Comments of Reviewer #3

Comments: The authors have followed the previous comments from the editor and reviewers to make revisions in accordance.

Response: We very much appreciate your recognition on our revised work. Following the comments from you, we have carefully revised the expression of hypothesis, clarified our empirical model and corrected typos. We are very sure that your comments have improved the quality of this paper greatly. 

Comments: The statements in hypotheses 1 to 3 are not consistent with the usual academic writing. In a hypothesis, there will not be the term ‘significantly’. Significance (or insignificance) is the result from statistical tests. Therefore, Hypothesis 1 may be like “CSR has a positive effect on cross-border M&A.” Usually there will not be a sentence of “That is,…” in a research hypothesis. Research hypotheses 2 and 3 are also strange because they are not expressed in complete sentences. They may be revised as “M positively (or negatively) moderates the effect of X on Y. 

Response: Thank you very much for your kind and detailed suggestions. Following your suggestions, we have revised the statements in hypotheses 1 to 3. 

Comments: However, the dataset are during 2010-2016. Tables 6-7 are results from fixed-effects panel regressions. Since each M&A is a one-shot event, maybe it is fine to use fixed-effects regression instead of random-effects data regression. The authors need to justify why a fixed-effects Logit regression is appropriate to use for the data analysis. If possible, some statistical tests should be provided to justify the use of fixed-effects regressions. 

Response: Thank you very much for your kind and constructive suggestions. We totally agree with you that the M&A is one-shot event, and our dataset is not a panel data although it is distributed from 2010 to 2017. Thus, the statement about the model we need to clarify first is that our empirical model is not a fixed-effect panel regression model but a pooled cross section model. We are very sorry about the inappropriate statements using the word of “fixed effect” in Table 6, Table 7, and others. We have revised all the statements in the manuscript and clarified that we just added the relevant dummies but not used the fixed-effect model.

Specifically, considering that some unobserved factors related with time, region, and industry may influence cross-border M&As, we generated firm, year, industry, and province dummies and added them into the model, thereby enhancing the model rigorousness and ensuring the estimation accuracy. This specification can also be seen in prior studies (e.g., Li et al., 2020; Gomes, 2019).

Thank you very much again for your careful and professional suggestions.

Comments: The current conclusion now can be directly supported from the empirical findings in this paper. The authors’ efforts in revising this paper should be positively affirmed. 

Response: Thank you very much for your positive remarks on our efforts. Your constructive suggestions greatly help us improve the quality of this paper. 

Comments: There are still some typos and format problems in the revised manuscript which need to be carefully corrected. For instance, in the Figure, H1, H2, and H3 cover parts of the arrows (paths). On page 38, it shows “42. 42.” 

Response: Thank you very much for your kind and constructive suggestions. Following your advice, we went through the whole paper very carefully and corrected the typos and problems. 

Thank you very much again! We greatly appreciate your help and suggestions. Hopefully, you will find our revisions satisfactory. 

References:

[1] Li WL, Wang CH, Ren QZ, Zhao D. Institutional distance and cross-border M&A performance: A dynamic perspective. Journal of International Financial Markets, Institutions and Money. 2020;66. doi: 10.1016/j.intfin.2020.101207. 

[2] Gomes M. Does CSR influence M&A target choices? Finance Research Letters. 2019; 30: 153–159. DOI:10.1016/J.FRL.2018.09.011.

---

## [Decision Letter · Decision Letter 3]

23 Dec 2021

Corporate Social Responsibility and Cross-border M&A: The Moderating Effect of Institutional Distance

PONE-D-21-18638R3

Dear Dr. Li,

We’re pleased to inform you that your manuscript has been judged scientifically suitable for publication and will be formally accepted for publication once it meets all outstanding technical requirements.

Kind regards,

Stefan Cristian Gherghina, PhD. Habil.

Academic Editor

PLOS ONE

Additional Editor Comments (optional):

Reviewers' comments:

Reviewer's Responses to Questions

**Comments to the Author**

1. If the authors have adequately addressed your comments raised in a previous round of review and you feel that this manuscript is now acceptable for publication, you may indicate that here to bypass the “Comments to the Author” section, enter your conflict of interest statement in the “Confidential to Editor” section, and submit your "Accept" recommendation.

Reviewer #3: All comments have been addressed

2. Is the manuscript technically sound, and do the data support the conclusions?

Reviewer #3: Yes

3. Has the statistical analysis been performed appropriately and rigorously? 

Reviewer #3: Yes

4. Have the authors made all data underlying the findings in their manuscript fully available?

Reviewer #3: Yes

5. Is the manuscript presented in an intelligible fashion and written in standard English?

Reviewer #3: Yes

6. Review Comments to the Author

Reviewer #3: All suggestions from this reviewer have been followed in accordance by the authors. I have no further suggestion.

7. PLOS authors have the option to publish the peer review history of their article (what does this mean?). If published, this will include your full peer review and any attached files.

Reviewer #3: No

---

## [Editor Report · Acceptance letter]

19 Jan 2022

PONE-D-21-18638R3 

Corporate social responsibility and cross-border M&A: The moderating effect of institutional distance 

Dear Dr. Li:

I'm pleased to inform you that your manuscript has been deemed suitable for publication in PLOS ONE. Congratulations! Your manuscript is now with our production department. 

Kind regards, 

on behalf of

Dr. Stefan Cristian Gherghina 

Academic Editor

PLOS ONE